# Olive Oil with Ozone-Modified Properties and Its Application

**DOI:** 10.3390/molecules26113074

**Published:** 2021-05-21

**Authors:** Marta Radzimierska-Kaźmierczak, Krzysztof Śmigielski, Magdalena Sikora, Adriana Nowak, Aleksandra Plucińska, Alina Kunicka-Styczyńska, Karolina H. Czarnecka-Chrebelska

**Affiliations:** 1Institute of Natural Products and Cosmetics, Lodz University of Technology, Stefanowskiego 4/10, 90-924 Lodz, Poland; magdalena.sikora@p.lodz.pl; 2Department of Environmental Biotechnology, Lodz University of Technology, Wolczanska 171/173, 90-924 Lodz, Poland; adriana.nowak@p.lodz.pl; 3Institute of Fermentation Technology and Microbiology, Lodz University of Technology, Wolczanska 171/173, 90-924 Lodz, Poland; aleksandra.plucinska@dokt.p.lodz.pl (A.P.); alina.kunicka@p.lodz.pl (A.K.-S.); 4Department of Biomedicine and Genetics, Chair of Biology and Medical Microbiology, Medical University of Lodz, 251 Pomorska Str., 92-213 Lodz, Poland; karolina.czarnecka@umed.lodz.pl

**Keywords:** ozonated olive oil, cosmetic emulsions, stability, antimicrobial activity, cytotoxicity

## Abstract

Olive oil application in the cosmetic industry may be extended by its ozonation, bringing about new oil properties and increased stability. Olive oil treated with 0.04 mole O_3_ or 0.10 mole O_3_ per 100 g oil was subjected to chemical parameters evaluation and composition scrutinizing by gas chromatography–mass spectrometry (GC-MS) and headspace solid-phase microextraction (HS-SPME) GC-MS analysis. The biological activity of refined and ozonated oil included their antimicrobial properties by the agar diffusion method and cytotoxicity by the MTT assay towards two normal (LLC-PK1, HaCaT) and two cancerous (Caco-2, HeLa) cell lines. The oils served as the basis in cosmetic emulsions. The chosen organoleptic features, preservative efficacy in a challenge test, and persistency during six months of these formulations were assessed. However, the ozonation of the olive oil resulted in a decrease in unsaturated acids; several additional compounds were detected in the ozonated oil, which positively affect the physicochemical, sensory, and functional properties of cosmetic emulsions. Emulsions based on the ozonated olive oil retain their properties longer compared to emulsions based on the refined olive oil. Ozonated oil treated with 0.10 mole O_3_/100 g oil allowed increasing the shelf life of the non-preserved formulation up to six months. A weak inhibitory effect against *Candida albicans* and *Aspergillus brasiliensis* was also demonstrated for this emulsion in the challenge test. Moreover, an interesting aroma, slightly enhanced antimicrobial activity against *Escherichia coli*, *Staphylococcus aureus*, *C. albicans*, *A. brasiliensis,* and a lack of cytotoxicity at concentrations 625 µg mL^−1^ make the ozonated olive oil a promising raw material for the cosmetics and pharmaceutical industries.

## 1. Introduction

Olive oil is a vegetable oil obtained by pressing of ripe olives. This raw material is often used in food and cosmetic industries due to its rich chemical composition and attractive utility and sensory features. The process of ozonation causes changes in the chemical composition of oil and brings about new biological properties. The addition of ozonated olive oil to cosmetic formulations enables producing innovative cosmetics, with novel, more attractive properties as it ensures a natural aroma and facilitates the penetration of active ingredients into the deeper parts of the skin, increasing the effect of hydration, and it may protect the cosmetics from microbial contamination. The majority of olive oil fatty acids are monounsaturated with oleic acid as predominant (65–85%). Except for fatty acids, olive oil is a source of bioactive compounds such as tocopherols or phenolic compounds [1]. The ozone mainly interacts with double carbon–carbon bonds in unsaturated fatty acids that yields peroxides, aldehydes, and ozonides, which are considered to be responsible for an enhancement of ozonated oil biological activity. These compounds’ amounts and ratio depend inter alia on the time of ozonation, the temperature of the process, and the type of reactor [2,3]. The ozonated olive oil is shown to express an antibacterial activity against *Staphylococcus aureus*, *Escherichia coli*, *Pseudomonas aeruginosa,* and *Bacillus subtilis*. Moreover, an increase in the antibacterial activity of ozonated oil with the increase of the peroxide number is observed [3]. Ozonated olive oil has already been successfully used in the treatment of several skin diseases and hypersensitivity, such as atopic dermatitis, contact dermatitis, ichthyosis, psoriasis, acne, chafes, pressure ulcers, blisters caused by insect bites, first-degree burns, a diabetic foot, as well as in skin care after laser therapy, surgery, and sunburn. It was found to be a good therapeutic agent in the treatment of asthma, gastrointestinal ulcers, and as a laxative in intestinal infections, including the ones caused by *Giardia lamblia*. Recent reports indicate that ozonated olive oil relieves in infections caused by human pinworms, genital herpes, human papillomavirus (HPV), and fungi *Candida* sp. [4]. It is also used in adjuvant treatment in orthopaedics and traumatology of organs movement. The ozonated olive oil is used topically on the skin [4,5] or as a component of formulations such as emulsions, ointments, and creams, which are easier to apply. The cosmetic industry increasingly uses the ozonated olive oil in creams and balms produced for the retail trade and the professional aesthetic medicine. High stability, a constant peroxide number, no trace of foreign smells, or changes in the structure for at least six months are the advantages of this oil over others. Furthermore, the production of ozone oil is neither expensive nor tedious [1,2,4,6,7].

The main aim of the research was to assess the usefulness of ozonated olive oil as a raw material for cosmetic emulsions. The GC-MS analysis was carried out to check the composition of fatty acids before and after oil ozonation. A broad evaluation of the biological activity of the refined and ozonated olive oils was conducted including their antimicrobial properties and cytotoxicity towards four cell lines (normal LLC-PK1, HaCaT and cancerous Caco-2, HeLa). The cosmetics preservative ability in the challenge test accompanied by the emulsions’ structure assessment was estimated.

## 2. Results and Discussion

### 2.1. Changes in the Chemical Composition of Olive Oil after Ozonation

Two batches of ozonated olive oil were prepared with the final ozone concentration 0.04 mole O_3_/100 g oil and 0.10 mole O_3_/100 g oil, corresponding to 2 g and 5 g of ozone in 100 g olive oil, respectively. To assess the chemical composition of the refined oil (oil before ozonation) and after ozonation, the analysis of fatty acid profile after esterification was performed. The refined olive oil was of a similar composition to the olive oils reported in the literature [8,9,10]. Differences in the composition of the oil before and after ozonation result from the mechanism of the process itself. Before the reaction, the main components of the oil were fatty acids, while after the passage of ozone, new compounds are formed, including those with shorter carbon chains. The transitions were realized by the disruption of fatty acid molecules at the site of double bonds, the attachment of oxygen atoms, the breakdown of these unstable intermediates, and the formation of oxygen derivatives of organic compounds with different properties. Reactions in the system involved both ozone and radicals. The data presented in Table 1 demonstrate the changes in the composition of the oil depending on the ozone concentration. The levels of hexadecanoic acid, (9*Z*)-hexadec-9-enoic acid, and octadecanoic acid were stable even after the introduction of 0.10 mole O_3_/ 100 g oil. Significant decreases in the content of the oleic acid and linoleic acid were observed in the oil after ozonation (in comparison to refined olive oil) (*p* = 0.0273 and *p* = 0.0273 respectively; Kruskal–Wallis test). Eight compounds were formed after 0.10 mole of ozone and five compounds after 0.04 mole of ozone treatment. Significant increases were observed in five compounds’ concentrations: nonanal, nonanoic acid, 9-oxonone acid, hexanal, and nonanedioic acid (*p* = 0.0241, *p* = 0.0241, *p* = 0.0241, *p* = 0.0241, *p* = 0.0241, respectively; Kruskal–Wallis test). Three compounds formed only in ozonated olive oil with 0.10 mole O_3_/100 g shown up in statistically insignificant amounts (*p* > 0.05; Kruskal–Wallis test). A decrease in the amount of oleic acid in the olive after ozonation and a concomitant increase in the contents of nonanal and nonanoic acid results from the reaction of ozone with the double bond in the fatty acid. Oxonone acid is the result of the action of an oxygen radical on the oleic acid molecule, and its amount increases with the concentration of ozone in the system (Figure 1 and Figure 2). The loss of unsaturated acid, expressed as (9*Z*)-octadec-9-enoic acid, was 12.3% and 29.1% at the 0.04 mole O_3_ and 0.10 mole O_3_/100 g, respectively. The low humidity of the olive oil results from the technological process conditions. According to the European Union Regulation, the moisture content in olive oil should not exceed 1% (Commission Regulation (EEC) No 2568/91) [11].

The ozonated olive oil was characterised by an interesting, quite strong odour. The compounds responsible for the oil’s flavour after the ozonation were identified by HS-SPME analysis (Figure 3). Interestingly, volatile compounds have not been detected in the headspace of olive oil. The odour of ozonated oils was mainly determined by aldehydes such as nonanal and hexanal (Table 2). In olive after ozonation with 0.10 mole O_3_/100 g oil, a statistically significant increase in the content of non-4-enal, nonanal, and hexanal were observed in comparison to the untreated oil (all *p* = 0.0241, *p* = 0.0429, *p* = 0.0429 respectively; Kruskal–Wallis test). An intensive aroma of the oil treated by the ozone with the higher concentration resulted in the higher content of nonanal (Figure 2 and Figure 3). Nonanal, at its highest level in the headspace, has a very strong, diffusive, fat-floral, waxy fragrance. At the appropriate dilution, the greasy notes become more pleasant, floral-waxy, more pink and sweet, fresh as neroli [12]. Hexanal, an aldehyde affecting the final aroma of the olives, has a very strong, penetrating, greasy, green, and grassy scent. At low concentrations, it resembles the freshly cut grass and immature fruits of apple and plum. A harsh note of highly concentrated material resembles rancid butter [12].

The interesting smell and potential useful properties of ozonated olive oil can bring about their application in cosmetics.

### 2.2. Density of Oils before and after Ozonation

The olive oils’ density after the ozonation differed statistically comparing to the untreated one. The increases in oil density after ozonation were statistically significant, in both 20 °C and 40 °C (both *p* = 0.0218, Kruskal–Wallis test). Density of oils decreases slightly with the temperature increase, (*p* < 0.05, Kruskal–Wallis test). Density values of ozonated and non-ozonated oils are shown in Table 3. The oil density is regarded as the parameter of high importance in the cosmetic production and compatibility with the other components of formulation. Moreover, the density of oil in the temperature of 40 °C can affect the rheology of cosmetic applied topically on human skin. The ozonated olive oil density values indicate that it is a favourable parameter in the formulation.

### 2.3. Characteristic Parameters of Oils before and after Ozonation

The characteristics of the oil significantly change with the ozonation process. The acid and peroxide values had a statistically significant increase after ozonation (*p* = 0.0338, *p* = 0.0218 respectively; Kruskal–Wallis test) (Table 4). The increase in the ozonated oil peroxide value indicates the presence of compounds with an oxidising potential. According to the literature data [2,3] and our results, ozonation increases the peroxide value with the amount of ozone introduced to the matrix. The iodine value decreased in the oil after the ozonation process (*p* = 0.0218, Kruskal–Wallis test), which is the effect of a decrease in the number of double bonds due to ozonolysis, resulting in the formation of organic compounds of a different structure [2,3]. In general, in comparison to the untreated oil, deeper changes in acid, iodine, and peroxide values were observed for oil after ozonation with 0.10 mole O_3_/100 g (Table 4).

### 2.4. Cytotoxicity of Ozonated Oils

Two normal (HaCaT and LLC-PK1) and cancerous (Caco-2 and HeLa) cell lines were challenged with refined and ozonated olive oil over a range of concentrations from 39 to 1250 µg mL^−1^. The dependence between the tested oil concentration and its cytotoxicity as well as statistical analysis (ANOVA, *p <* 0.01) are presented in a tabular form (Table 5). Refined oil displayed no cytotoxicity (*p <* 0.01) towards three cell lines (LLC-PK1, Caco-2, and HeLa). The only slight cytotoxic effect of the HaCaT cells was observed for oil in concentrations 312 and 1250 µg mL^−1^ (*p* < 0.01).

The increase in cytotoxicity was not linear, but it persisted at the similar level. Non-cancerous cell lines (HaCaT and LLC-PK1) seemed to demonstrate slightly higher sensitivity to ozonated oil (*p <* 0.01) than cancerous ones (HeLa and Caco-2) (*p <* 0.01), but generally, the oil demonstrated slight cytotoxicity—up to approximately 32 and 36% in the highest concentration for HaCaT and LLC-PK1, respectively (Table 5). The viability of HeLa and Caco-2 cancerous cell lines was persistent for all tested oil concentrations (above 80%). In addition, there were no statistically significant differences (*p <* 0.01) between cytotoxicity induced by both ozonated oils (0.04 mole O_3_ and 0.10 mole O_3_/100 g), so it did not depend on the degree of ozonation. When comparing the cytotoxicity of refined and ozonated oils at the same concentrations for each cell line, results were not significantly different; ANOVA (*p <* 0.01).

Quantitative evaluation according to ISO 10993-5 standard [9] of the cytotoxic effects of ozonated oils showed slight (i.e., not more than 20% of the cells demonstrate changes in morphology or growth inhibition) or no cytotoxicity (i.e., discrete or no changes in morphology) on cells at the indicated concentrations in comparison to the tests on cells not subjected to the ozonated oils treatment. For cancerous cell lines (Caco-2 and HeLa), no morphological changes were observed. The ozonated oils at the highest concentrations tested (625 and 1250 µg mL^−1^) against normal cells (LLC-PK1 and HaCaT, respectively) expressed mild cytotoxicity (i.e., not more than 50% of the cells demonstrate changes in morphology or growth inhibition). It follows from the above that oils at concentrations below 625 µg mL^−1^ appear safe for application. Nevertheless, cellular models in vitro are not equivalent to in vivo tests, and they represent only a predictive model.

### 2.5. Antimicrobial Activity of Olive Oil

According to the scale adapted [13], the refined olive oil was classified as active against *P. aeruginosa* and *B. subtilis* with slightly higher but statistically significant (*p* < 0.05) activity towards *B. subtilis* (Figure 4). The intact olive oil did not affect the growth of bacteria *S. aureus* and *E. coli*, yeast *C. albicans*, and moulds *A. brasiliensis* (no statistically significant differences). The ozone treatment in the dose of 0.04 mole O_3_/100 g oil slightly enhanced the antimicrobial activity of the olive oil against the tested microorganisms excluding *P. aeruginosa* and *B. subtilis*, resulting in the statistically significant increase in the growth inhibition zones up to 9–11.5 mm. The highest dose of ozone did not boost the antimicrobial action of the olive oil with the exception of *C. albicans*. The only olive oil with 0.04 mole O_3_/100 g oil ozone treatment was rated as active against all the tested microorganisms. However, the variety of ozonated plant oils was proved to express an antibacterial effect against *S. aureus*, *Enterococcus faecalis*, *Enterococcus faecium*, *Streptococcus pyogenes*, *E. coli*, *P. aeruginosa*, and *Mycobacterium* spp.; the majority of research concerns sunflower oil and its commercial preparations [14,15,16]. The ozonated olive oil was noted to be effective in in vivo studies in rats against *S. pyogenes* and *S. aureus* [17]. Our studies also revealed the highest activity of 0.04 mole O_3_/100 g ozonated olive oil against *S. aureus*. The ozonated olive oil is known to be effective against *C. albicans* in the treatment of vaginal mucosa [18,19,20], which is in agreement with the presented activity against *C. albicans* of both olive oils treated by ozone in different doses in in vitro tests. The olive oil after ozone treatment was noted to express antifungal activity against *Aspergillus fumigatus* and other dermatophytes [21], which is consistent with the proven statistically significant inhibitory effect of ozonated (0.04 mole O_3_/100 g) olive oil on *A. brasiliensis* in our research.

### 2.6. Cosmetic Emulsions

The refined and ozonated olive oil were the main components of the cosmetic emulsions. Structures of the emulsions differed depending on the oil applied (Figure 5). The less homogenous was the one based on the refined olive oil (Figure 5a). Application of the both ozonated olive oils tested increased the formulation homogeneity. The emulsion based on the ozonated olive oil with the highest ozone treatment (0.10 g O_3_/100 g oil) was noted as the most homogenous (Figure 5c).

Preliminary studies on a group of volunteers testing cosmetic preparations based on the ozonated olive oil showed the increased flexibility and evident epidermis hydration feeling. After application, the emulsions gave a silky feeling, smoothed the skin, and presented no stickiness effect. Moreover, users appreciated the interesting, fresh smell of the formulations.

### 2.7. Preservative Efficacy of Cosmetic Emulsions

The cosmetic emulsions formulated on the basis of the refined and ozonated olive oils were subjected to the challenge tests [22]. To check the preservative efficacy of the ozonated olive in the formulation, no synthetic preservatives were added. The survival of *E. coli* and *P. aeruginosa* was not suppressed during 28 days of incubation in any of the formulations irrespective of the olive oil treatment (Figure 6a,b). No statistically significant differences were observed among the formulations with olive oils. In the emulsions with the refined oil and the one treated by 0.04 mole O_3_/100 g olive, the number of *S. aureus* viable cells was gradually declining, reaching the maximum drop by about 3 log units at the end of incubation period (Figure 6c). The preservative activity of the emulsion with olive oil ozonated 0.04 mole O_3_/100 g was the strongest within two weeks of incubation, reducing the staphylococci level by 2 logarithmic units CFU. All the tested emulsions promoted the growth of *C. albicans* by 0.4–1.06 logarithmic units CFU up to 14 days (Figure 6d). After the next two weeks, the number of viable yeasts decreased by 1.9 logarithmic units in the emulsion with the olive oil treated by 0.10 mole O_3_/100 g. All the emulsion formulations enabled the growth of moulds *A. brasiliensis* (increase by 1.5–1.9 logarithmic units CFU) after 21 days of incubation. Extending the incubation time for the next 7 days resulted in the inhibition of the moulds growth by 1.9 log units in the emulsion with olive ozonated with 0.10 mole O_3_/100 g only (Figure 6e).

The literature data of preservative efficacy of cosmetics containing olive oil are scarce and primarily focussing on the formulation’s biophysical and microbiological stability [23]. A variety of beneficial actions of olive oil topically applied on a skin were shown in in vivo experiments, and the researchers are rather focusing on the cosmetics’ cutaneous effects [24,25,26]. However, the olea ointment with the olive oil as one of the main compounds was proved to prevent burns infections [27], indicating its antibacterial activity. Although the emulsions tested by us did not fulfil either A or B criteria for cosmetics’ preservative efficacy, the emulsion with olive ozonated by 0.04 mole O_3_/100 g expressed vivid anti-staphylococcal activity. This finding is in agreement with the antimicrobial activity tests of olive oils, where the 0.04 mole O_3_/100 g ozonated olive oil showed the statistically significant (*p* < 0.05) increase in anti-staphylococcal activity compared to the other olive specimens. The weak inhibitory effect against *C. albicans* and *A. brasiliensis* was also demonstrated for the emulsion with the addition of olive oil with the highest dose of ozone. Due to the lack of sufficient auto-preservative abilities of the emulsions with refined and ozonated olive oil, the supplementation of the formulations with synthetic preservatives will be considered and checked in further research.

### 2.8. Persistency of Cosmetic Emulsions

For microbiological stability testing, the emulsions were stored in an ambient temperature (about 20 °C) during six months. The formulations showed differences in their persistency (Figure 7). After one month of storage, the growth of *Aspergillus* sp. moulds on the surface of the cosmetic based on the refined olive oil was observed, and three out of four test samples were contaminated (Figure 7a). The visible growth of this mould was also noted on the surface of one test sample of the emulsion with olive oil treated with the lower dose of ozone (0.04 mole O_3_/100 g oil). The most stable emulsion was the one based on the ozonated olive oil with 0.10 mole O_3_/100 g oil (Figure 7c), showing no microbial contamination after six months of storage. The results of the persistency testing were in agreement with the challenge test results of weak inhibitory effect against *A. brasiliensis* in the formulation with the highest olive oil ozonation (Figure 6e). This microbiological stability of the cosmetic may be attributed to the ozonated olive oil with 0.10 mole O_3_/100 g oil.

## 3. Materials and Methods

### 3.1. Process of Ozonation

The raw material for ozonation was refined olive oil (Pol-skór, Łodź, Poland). Samples (100 g) of the oil were ozonated using gaseous oxygen containing 100 g ozone in m^3^, which was fed from the bottom of the reactor through a sintered glass. The rating flow of gaseous oxygen containing ozone was 0.5 dm^3^ min^−1^; pressure: 800 hPa. Each batch consisted of three samples. The overall ozone amounts were as follows: 2 g of ozone in 100 g olive oil corresponding to 0.04 mole ozone/100 g oil, time of ozonation 40 min; 5 g of ozone in 100 g olive oil corresponding to 0.10 mole ozone/100 g oil, time of ozonation 100 min. Ozonizer used in experiments was the apparatus designed in The Institute of Natural Products and Cosmetics at Lodz University of Technology (Scheme 1).

### 3.2. Esterification with Derivatisation for the Analysis of the Composition of Oils

To carry out the esterification with derivatisation, 20 μL of the sample, 400 μL tert-butyl methyl ether (99.5%, Sigma-Aldrich, St. Louis, MO, USA), and 100 μL TMSH (Fluka Analytical, Munich, Germany) were used. The samples were incubated at 60 °C for 30 min. Each experiment was conducted in triplicate.

### 3.3. GC-MS Analysis of Oil before and after the Ozonation Process

GC-MS analyses were carried out using a Trace GC Ultra gas chromatograph coupled with a DSQ II mass spectrometer (Thermo Electron, Waltham, MA, USA) and equipped with a STABILWAX^®^-DA (Restek, Bellefonte, PA, USA) (30 m × 0.18 mm × 0.18 μm) column. Before the analysis, the samples were subjected to esterification with derivatisation. The GC-MS conditions were as follows: injection temperature: 50 °C, detector temperature: 240 °C, developing gas: He pressure: 110 kPa, split: 50 cm^3^ min^−1^, temperature program: 50 °C (2 min), 10 °C min^−1^ (230 °C), 230 °C (30 min). Each sample was conducted in triplicate.

Database to GC-MS analysis: The NIST Library, Wiley 8th edition, Adams 4th edition.

### 3.4. Analysis of the Ozonated Oil Headspace Phase by HS-SPME

To examine the composition of the phase above the surface of the oil, 3.0 g of the sample was incubated at 40 °C for 20 min. Each batch consisted of three samples. HS-SPME analysis was carried out using a ternary fibre Divinylbenzene/Carboxen/Polydimethylsiloxane (DVB/CAR/PDMS). Extraction time: 10 min, temperature: 40 °C. Gas Chromatography was conducted using a Trace GC Ultra chromatograph coupled with a DSQ II mass spectrometer (Thermo Electron, Waltham, MA, USA) and equipped with a STABILWAX^®^-DA (30 m × 0.18 mm × 0.18 μm) column. The HS-SPME conditions were as follows: injection temperature: 50 °C, detector temperature: 240 °C, developing gas: He, pressure: 110 kPa, split: 50 cm^3^ min^−1^, temperature program: 50 °C (2 min), 10 °C min^−1^ (230 °C), and 230 °C (30 min). Each experiment was conducted in triplicate.

### 3.5. Density Determination

Density determination of the oils was conducted using an Automatic Density Meter DDM 2910 (Rudolph Research Analytical, Hackettstown, NJ, USA).

### 3.6. Determination of Acid, Peroxide, and Iodine Values

The acid number, peroxide number, and iodine number were determined in accordance with applicable PN-ISO standards [28,29,30]. Each experiment was conducted in triplicate.

Acid value: First, 10 g of the oil sample was dissolved in an ethanol/ethyl ether (1:1) mixture (Sigma-Aldrich, St. Louis, MO, USA); 0.1 N KOH (Sigma-Aldrich, St. Louis, MO, USA) was used for titration. The blank sample was alike but without the oil. The acid value (AV) was determined according to Equation (1).
(1)$$AV=\frac{56.1\cdot \left(V-V_{0}\right)\cdot c}{m}\;\left[\frac{mg_{KOH}}{g}\right],$$

*V*—volume of 0.1 N KOH used for oil sample titration; *V*_0_—volume of 0.1 N KOH used for blank sample titration; *c*—KOH concentration (0.1 N); *m*—mass of oil sample; gram (*g)*.

Peroxide value: First, 2 g of oil sample was dissolved in 10 mL of chloroform (Sigma-Aldrich, St. Louis, MO, USA). Then, 15 mL of acetic acid (Sigma-Aldrich, St. Louis, MO, USA) and 1 mL of potassium iodide solution (Sigma-Aldrich, St. Louis, MO, USA) were added, mixed, and kept in darkness for 5 min. After incubation, 75 mL of distilled water was added, and the sample was titrated with 0.002 N sodium thiosulphate solution (Sigma-Aldrich, St. Louis, MO, USA). The blank sample was alike but without the oil. The peroxide value (PV) was determined according to Equation (2).
(2)$$PV=\frac{\left(V-V_{0}\right)\cdot 0.002}{m}\cdot 1000\;\left[\frac{mEqO}{kg}\right],$$

*V*—volume of 0.002 N sodium thiosulphate used for oil sample titration; *V*_0_—volume of 0.002 N sodium thiosulphate solution used for blank sample titration; *m*—mass of oil sample; milliequivalents of active oxygen (*mEqO)*.

Iodine value: First, 0.13 g of oil sample was dissolved in 20 mL of cyclohexene/acetic acid (1:1) mixture (Sigma Aldrich, St. Louis, MO, USA), and 25 mL of Wijs solution (Sigma-Aldrich, St. Louis, MO, USA) was added, mixed, and left for 1 h in darkness. Afterwards, 20 mL of potassium iodide solution and 150 mL of distilled water were added. Then, 0.1 N sodium thiosulphate (Sigma-Aldrich, St. Louis, MO, USA) was used for titration. The blank sample was alike but without the oil. The iodine number (IV) was determined according to Equation (3).
(3)$$IV=\frac{\left(V_{0}-V_{1}\right)\cdot N\cdot 12.69}{m}\;\left[\frac{g_{Iodine}}{\;100\;g}\right],$$

*V*_0_—volume of 0.1 N sodium thiosulphate used for blank sample titration; *V*_1_—volume of 0.1 N sodium thiosulphate solution used for oil sample titration; *N*—normality of sodium thiosulphate solution (0.1 N); *m*—mass of oil sample; gram of iodine (*g_Iodine_)*.

### 3.7. Cytotoxicity Testing

#### 3.7.1. Chemicals and Reagents

HEPES buffer, Dulbecco’s Modified Eagle’s Medium (DMEM), DMEM:Ham’s F12 medium, streptomycin/penicillin mixture, phosphate-buffered saline (PBS, pH 7.2), trypan blue dye, 3-(4,5-dimethylthiazol-2-yl)-2,5-diphenyltetrazolium bromide (MTT), dimethyl sulfoxide (DMSO) derived from Sigma-Aldrich (St. Louis, MO, USA). Foetal bovine serum (FBS), GlutaMAX^TM^, TrypLE^TM^ Express originated from Invitrogen Thermo Fisher Scientific (USA); 0.22 μm pore size syringe filters were from Merck Millipore (Germany).

#### 3.7.2. Stock Solutions Preparation of Oils

The stock solutions of ozonated and refined oils were freshly prepared after dissolving in 5% ethanol in PBS. They were sterile-filtered (0.22 μm pore size), diluted in PBS to the 10× stock concentrations, and stored at room temperature. The maximum final concentration of ethanol for which cells were exposed to was 0.5%, which was not toxic for cell lines tested (unpublished data).

#### 3.7.3. Cell Culturing

In the research, two normal (LLC-PK1 and HaCaT) and two cancerous (Caco-2 and HeLa) cell lines were applied. Human colon adenocarcinoma cell line Caco-2 (41st passage), swine kidney epithelial cell line LLC-PK1 (38th passage), and human keratinocyte cell line HaCaT (31st passage) were purchased from Cell Line Service GmbH (Eppelheim, Germany). HaCaT is the original material created by Prof. Dr. Petra Boukamp and Dr. Norbert Fusenig [31]. The human cervix adenocarcinoma cell line HeLa was a gift from a professor who wanted to stay anonymous. Caco-2, HeLa, and HaCaT cells were cultured in T75 flasks in DMEM, supplemented with 10% FBS, 4 mM GlutaMAX^TM^, 25 mM HEPES buffer, and 100 µg/mL streptomycin/100 IU/mL penicillin mixture for 5–7 days at 37 °C in the atmosphere of 5% CO_2_. In the case of LLC-PK1, DMEM/Ham’s F12 medium and 5% FBS were used. Every 2–3 days, cells were washed with PBS and the medium was renewed. Confluent cells were detached with TrypLE^TM^ Express for 5–10 min, depending on the cell line. The cell suspensions were centrifuged (182× *g*, 3–5 min), decanted, and the pellets were re-suspended in fresh DMEM. After determination of cell count by haemocytometer and cell viability by trypan blue exclusion, the cells were ready to use.

#### 3.7.4. MTT Assay

Cytotoxicity was evaluated quantitatively by a reduction of MTT (3-(4,5-dimethylthiazol-2-yl)-2,5-diphenyltetrazolium bromide) assay. First, 1 × 10^4^ cells in the logarithmic growth phase were seeded in each well of a 96-well plate in complete culture medium and incubated overnight (37 °C, 5% CO_2_). Next, the medium was aspirated, and tested oils were added to achieve final concentrations from 39 to 1250 µg mL^−1^. The negative control contained cells in DMEM. Cells were exposed to extracts for 24 h (37 °C, 5% CO_2_). After incubation, the samples were aspirated, MTT (0.5 mg mL^−1^ in PBS) was added, and they were incubated for further 3 h. Next, it was removed and formazan precipitates were solubilised by DMSO. Absorbance was measured at 550 nm (with a reference filter of 620 nm) using a microplate reader (TriStar^2^ LB 942, Berthold Technologies GmbH & Co. KG, Germany).

The absorbance of the control sample (untreated cells) represented 100% cell viability. Cell viability (%) was calculated as: (sample OD/control OD) × 100%; and cytotoxicity (%) was calculated as: 100—cell viability. Results were presented as mean of four individual evaluations ± standard deviation (± SD). Cytotoxicity responses were evaluated according to ISO 10993-5:2009(E) [9]. The mean error of MTT method is up to 10%.

### 3.8. Cosmetic Formulation

The compounds used to make the oil-in-water emulsion were selected to act synergistically. The methodology of obtaining the emulsions was patented [32,33,34]. The formulation ingredients are presented in Table 6.

### 3.9. Microorganisms

The following strains were used during the experiments: Gram-negative bacteria *E. coli* ATCC 8739, *P. aeruginosa* ATCC 9027; Gram-positive bacteria *S. aureus* ATCC 6538, *B. subtilis* ATCC 6633; yeast *C. albicans* ATCC 10231 and moulds *A. brasiliensis* ATCC 1640. All the microorganisms were the reference strains originated from the American Type Culture Collection ATCC. The bacteria were cultured on Trypticase Soy Agar (TSA, Oxoid, Basingstoke, UK) and yeast and moulds were cultured on Sabouraud Dextrose Agar (SDA, bioMerieux, Warsaw, Poland). All the microorganisms were twice subcultured in the respective media before each experiment. The broths of inoculated *E. coli*, *S. aureus,* and *P. aeruginosa* at 37 °C were incubated for 24 h, *B. subtilis* was incubated at 30 °C for 24 h, and yeast and moulds were incubated at 25 °C for 24–72 h. The physiological salt solution (0.85% NaCl) was used for inoculum preparation of each strain.

### 3.10. Determination of Antimicrobial Activity of Olive Oil

The agar disc diffusion method was used to estimate the antimicrobial activity of the refined and ozonated olive oils. Sterile paper discs of 6 mm in diameter (Whatman No. 40, Lab-System-Service, Szczecin, Poland) were soaked with the oil. The discs were placed onto the surface of the appropriate medium previously inoculated with the specific strain of the microorganisms. The media, TSA and SDA for bacteria and fungi, respectively, assuring the optimal growth of the microorganisms were applied. The amount of 0.1 mL of the standardised suspensions (10^8^ CFU mL^−1^) of the tested microorganisms was transferred onto the agar medium. Petri dishes were kept at 4 °C for 2 h and then incubated at the temperatures optimal for specific microorganisms for 24–72 h, and the zones of inhibition were measured. The antimicrobial activity of the olive oil samples against the specific microorganisms was classified by the diameter of the inhibition zones as follows: not active for diameter less than 8 mm, active for diameter 9–14 mm, very active for diameter 15–19 mm [13]. Results were presented as an arithmetic mean of six determinations with standard deviation and were analysed using a 3-way ANOVA test at a confidence level of *p* < 0.05.

### 3.11. Challenge Test

Challenge tests of cosmetics were conducted according to ISO 11930:2012 regulations in compliance with European Pharmacopoeia 7.0. The amount of 1 mL suspension of active bacteria (10^7^–10^8^ CFU mL^−1^) or yeast (10^6^–10^7^ CFU mL^−1^) in a physiological salt solution was introduced to 100 mL of the cosmetic. The mould suspension (10^6^–10^7^ CFU mL^−1^) was prepared in physiological salt solution with the addition of 0.5 g L^−1^ polysorbate 80. After thorough mixing, the cosmetics were incubated in the dark at 20 °C. The count plate method was used for determination of the number of viable cells in formulations immediately after inoculation and 2, 7, 14, and 28 days. A sample of 10 mL cosmetic was transferred to 90 mL suspending liquid (peptone K 5.0 g L^−1^, NaCl 8.5 g L^−1^, pH 7.2 ± 0.2) and inoculated on TSA (bacteria) and SDA plates (fungi). The plates were incubated at 37 °C and 28 °C for bacteria and fungi, respectively. The results were expressed as log CFU mL^−1^.

The cosmetics were evaluated according to the criteria of the acceptance ISO for topical preparations [22]. Criterion A for bacteria was fulfilled at a reduction of the microorganism inoculum by 3 logarithmic units within 7 and 14 days of the challenge test with no increase up to the 28th day. Criterion A for yeasts is a reduction of the inoculum by 1 logarithmic unit within 7 days with no increase up to the 28th day, and for moulds, there was no increase within 14 days and a reduction by 1 logarithmic unit within 28 days. Criterion B for bacteria was achieved by a reduction of the inoculum by 3 logarithmic units within 7 and 14 days of a challenge test with no increase up to the 28th day; for yeasts, there was a reduction of the inoculum by 1 logarithmic unit within 7 days with no increase up to the 28th day; and for moulds, there was no increase of inoculum within 28 days. Results were presented as an arithmetic mean of three determinations with standard deviation and were analysed using a three-way ANOVA test at a confidence level of *p* < 0.05.

### 3.12. Cosmetic Persistency Testing

The microbiological persistency of the cosmetics was evaluated during six months of cosmetics storage in the ambient temperature about 20 °C. Every two weeks, the cosmetics were scrutinised in aseptic conditions to check the microbiological contaminations. In case of microbial growth, the macroscopic and microscopic observations of the microorganisms growing onto the sample surfaces were conducted. The four cosmetic emulsions samples based on each olive oil tested were stored in tightly closed containers.

### 3.13. Statistical Analysis

The results were presented as mean value ± SD. Differences regarding variables of the olive oil; i.e., density and characteristic numbers (i.e., acid number, peroxide number, iodine number) before and after the ozonation process were tested using the Kruskal–Wallis test (KWW test), followed by multiple comparison test (MCT) to indicate significant differences between the groups at *p* < 0.05. Cytotoxicity assay data were analysed using one-way analysis of variance (ANOVA). The significant differences between the means were compared using Scheffe’s multiple comparison test at *p <* 0.01. The results of antimicrobial activities were analysed using a 3-way ANOVA test at a confidence level of *p* < 0.05. All statistical analyses were performed by means of Statistica ver. 13.1 (StatSoft, Tulsa, Oklahoma, USA).

## 4. Conclusions

The ozonation of the olive oil resulted in changes in its chemical composition. Although a decrease in unsaturated acids was observed, several additional compounds detected in the ozonated olive oil positively affect physicochemical, sensory, and functional properties of cosmetic emulsions. Emulsions based on the ozonated olive oil retain their properties much longer compared to emulsions based on the refined olive oil. Ozonated olive oil treated with 0.10 mole O_3_/100 g oil allowed increasing the shelf life of the non-preserved formulation up to six months. A weak inhibitory effect against *C. albicans* and *A. brasiliensis* was also demonstrated for this emulsion in the challenge test. An interesting aroma and lack of cytotoxicity at concentrations 625 µg mL^−1^ make the ozonated olive oil a promising raw material for the cosmetics and pharmaceutical industries.

## Data Availability

The data presented in this study are available on request from the corresponding author.

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
