# Peer review of "Olive Oil with Ozone-Modified Properties and Its Application"

_molecules, 2021, doi:10.3390/molecules26113074_

Round 1

Reviewer 1 Report

I read the resubmitted version of the original paper entitled “Olive oil with ozone-modified properties and its application” from Radzimierska- KaĹşmierczak et al.

Once again, the paper has improvements in several ways. However, once again, the same original doubt remains in me. I still do not appreciate the real relevance of such a research in consideration of the characteristics of the selected Journal. In fact, “the chemical part is either incomplete or not very innovative or presents aspects that are not entirely correct”. Moreover, I still doubt about the full competency of the Authors about some aspects.

To better highlight these aspects, I enclose the revised paper with the questioning parts highlighted in yellow.

In detail, about the current peroxide number evaluation (see both page 6 of 17 and 12 of 17), I can basically report what I have already been stated in the previous revision:

The determination of the peroxide value (PV) is a highly empirical procedure. The selected bibliographic reference (ref. #29) dates back to 1996. Today, the ISO 3960:2017 “Animal and vegetable fats and oils — Determination of peroxide value — Iodometric (visual) endpoint determination” is the current version (cfr https://www.iso.org/standard/71268.html). In the abstract of this document you can find: “The method is applicable to all animal and vegetable fats and oils, fatty acids and their mixtures with peroxide values from 0 meq to 30 meq (milliequivalents) of active oxygen per kilogram”. In fact, the formation of free peroxides involves rancidity of the oils and is obviously to be avoided. On the contrary, the case in question is fully different. In fact, the oils at the level of the double bonds are expressly transformed into cyclic peroxides by the ozonation process. To date, to my knowledge and despite continuous research in this regard, there are still no internationally recognized standardized methods for the 1,2,4-trioxolane moieties quantitative evaluation (cfr. https://doi.org/10.1080/01919512.2020.1868972). On the other hand, based on the values shown in Table 4, the titration with 0.002 N sodium thiosulphate would have led to a dripping of a huge amount of titrant solution. In fact, assuming negligible the volume of titrant used for blank sample titration, the volume, in mL, of sodium thiosulphate used for oil sample titration is practically equal to the peroxide number value (ie more than 900 mL for ozonated olive oil treated with 0.10 mole O3/100 g).

Furthermore, to such a significant variation in the quantity of peroxides, unusually does not correspond to an analogous abatement of the iodine value (cf., again, https://doi.org/10.1080/01919512.2020.1868972). Among other things, the values found are not in line with those normally indicated for olive oil in the literature (cf. https://doi.org/10.1590/S0103-50532006000200026).

As for the process of ozonation (page 11 of 17), I recommend to represent the real operating conditions taking a cue from what was reported by one of the authors (K.Ĺš.) in a recent paper (cf. https://doi.org/10.3390/ijerph14101196). Moreover, if I correctly interpret the authors' sense, in my opinion the term “final ozone concentration” is inappropriate. In fact, it represents the “overall ozone amount” which came into contact in the reactor with the oil to be subjected to the ozonation process at the experimental conditions adopted. Among other things, based on the feeding gas flow and concentration data reported (100 g m-3 and 0.6 L min-1, respectively), I am not sure of the correctness of both 2 and 5 g of ozone in 100 g olive oil. Are they 2.4 g and 6 g, corresponding to 0.05 and 0.125 mol of ozone, respectively?

A further criticality emerges from the evaluation of the statistical analysis. In my opinion, it is highly unlikely to find identical p values by comparing different starting data both in terms of average values and in terms of SD.

Reviewer 2 Report

Dear Authors,

The manuscript is well designed and contains relevant information for the use of ozone, as innovative pre-treatment technology, in the production of cosmetic components. I would like to suggest few changes.

Line “is used topically directly on the skin” should be “is used topically on the skin”

Line “are this oil other advantages” should be “are the advantages of this oil other”

Please check the format of Table 6.

Author Response

Response to Reviewer 2 Comments

Authors are thankful for all the remarks concerning our manuscript. Please find below our explanation and corrections.

Point 1 Line “is used topically directly on the skin” should be “is used topically on the skin”

Response 1: The sentence was changed for “The ozonated olive oil is used topically on the skin…”

Point 2 Line “are this oil other advantages” should be “are the advantages of this oil other”

Response 2 We changed the sentence as follows “…or changes in the structure for at least six months are the advantages of this oil other.”

Point 3 Please check the format of Table 6.

Response 3 The format of Table 6 was changed and adjusted to the text formatting.

This manuscript is a resubmission of an earlier submission. The following is a list of the peer review reports and author responses from that submission.

Round 1

Reviewer 1 Report

Dear Authors,

The manuscript seems to present interesting results for the preparation of a cosmetic using ozonated olive oil. In general, the manuscript is interesting and well-written but more information is necessary to clarify the impact of the treatments.

Material and methods. Please, provide more information about the replications of the experiment, the quantity prepared for each batch, and the number of samples for each assay.

Please, revise the section 3.2. The language in this section is not appropriate.

Statistical analysis is missing and, consequently, the results and discussion and conclusion cannot be critically assessed without a proper indication of significant differences found (or not) in the results. Please, evaluate the pertinent results by a proper statistical test and make the necessary modifications in the results, discussion, conclusion and abstract of the manuscript.

Reviewer 2 Report

I read with both care and interest the original paper from Radzimierska-KaĹşmierczak et al. entitled “Olive oil with ozone-modified properties and its application”. The AA. performed a research “to obtain and assess the ozonized olive oil as a raw material used in cosmetic emulsions”.

The main observations that came to my mind when reading the manuscript are the following:

- First of all, the choice of some of the initial bibliographic references appears unusual. In fact, reff. #1-2 refer to two papers of little international importance, in Polish language. In reality, wanting to enter immediately into the specific use of olive oil and its ozonated derivatives, for years there have been some publications in specific journals of international relevance, like, eg:

https://www.tandfonline.com/doi/abs/10.1080/01919510590945822

https://www.tandfonline.com/doi/abs/10.1080/01919512.2014.904736

https://www.tandfonline.com/doi/abs/10.1080/01919512.2017.1341832

A specific book chapter has also recently appeared:

https://www.intechopen.com/books/herbal-medicine/powerful-properties-of-ozonated-extra-virgin-olive-oil

In my opinion, the content of these publications significantly resizes the relevance of the manuscript under review. However, regardless of a judgment on the merits, I believe that they must still be adequately taken into consideration and accordingly cited in the Introduction section. I also propose to replace the current ref #5 with a more visible one.

- The data presented will be more understandable separating the results and discussions sections.

- It is the first time that I hear it defined on an olfactory level the ozonized olive oil as “interesting, quite strong odor”! According to practically unanimous opinions, the ozonation process gives the vegetable matrices an unpleasant, pungent scent. This specific point opens up a very important question regarding the methods of gaseous ozone treatment. In fact, in the 3.1. Process of ozonation section, the parameters necessary for reproducibility of the preparation are not indicated at all. The only feed concentration of 100 g ozone/m3 does not allow going back to the final ozone concentrations in olive oil. In fact, it is not a solubilization process. As correctly indicated in other sections of the manuscript, the ozonation process involves formation of oxygen derivatives, mainly 1,2,4-trioxolane rings, at the site of double bonds. Gas flow, geometric characteristics of the reactor, reaction time and temperature, contact surface of the sintered glass are just some of the parameters to be kept under control.

- The section “2.4. Characteristics numbers of oils before and after ozonation” does not match in the part 3. Materials and Methods. Therefore, it is not possible to trace the methods used to obtain the values indicated in Table 4. This aspect is of particular importance, also in relation to the current lack of standardized methods for this type of products (cfr. https://pubmed.ncbi.nlm.nih.gov/18679737/;

https://pubmed.ncbi.nlm.nih.gov/19900426/;

https://www.tandfonline.com/doi/abs/10.1080/01919512.2020.1868972).

- The section “3.7. Receipt of cosmetic formulation” it cannot be limited only to the qualitative indication of the components, as indicated in Table 5. In fact, formulation parameters (surfactant-to-oil ratio, surfactant concentration, mixing rate, temperature, just to name a few) significantly affect preparation properties like mean particle diameters, size distributions, creaming. On the other hand, if a process patent protects the obtaining of the formulation, it would be advisable to indicate this in advance in the text.  Anyway, the term “oily-oil-based formulation” I think it is wrong.

Minor concerns

- The indication of the email of one corresponding author (M.R.-K.) is not reported.

- The bibliographic references order [14-16] is not consistent with the previous ones, where the last reference number is #7. In the same page 2 of 14, “he challenge test” should be corrected. Also the indication of bibliographic references [10-12] of Table 5 (page 12 of 14) is not consistent with the previous ones.

- Since the term "ozonation" is correctly indicated in place of “ozonization”, “ozonated oils” should be preferred to “ozonized oils”.

- Some compounds of Table 1 are indicated by unorthodox chemical terms

- Apart from what I have already stated regarding the absence of the quantification method of the final ozone concentrations, sometimes in the text the ozone content is different from those indicated in section 3.1 (cf. legends of Figures 5-8, with 4 g O3/100 g oil instead of 5 g O3/100 g oil)

- Page 10 of 14: the film thicknesses measure unit of STABILWAX®-DA (30 m x 0.18 mm x 0.18 Ä›m) column should be amended.

Round 2

Reviewer 1 Report

Dear Authors,

Thank you so much for providing additional information and improving the original manuscript.

I would like to reiterate that statistical analysis must be included in the manuscript in a proper way. In the case my comment was not clear in the first round, my apologies.

Please create section in the manuscript ate the end of material and method (section 3.12 Statistical Analysis) where the information regarding the information of replications of the experiment, the number of samples is presented and software used.

Please provide the statistical analysis for the results presented in the Tables 2, 3 and 4 and Figure 4 (sections 2.3, 2.4, 2.5, and 2.6) and also improve the discussion of results.

For instance, Section 2.4 (Characteristic numbers of oils before and after ozonation) provides information about the effect of ozonation in the acid number, peroxide value and iodine value. It is not clear how ozonation can affect these indicators because statistical analysis is missing, which rises questions like: Does ozonation increase, decrease or has no effect in these variables? The effect is in accordance with data available in the literature? Which other studies obtained similar/different results? Why?

I believe these questions must be considered to improve the interpretation and discussion of the results in sections aforementioned and provide a critical and scientific view of the manuscript as a whole.

Reviewer 2 Report

I still have some doubts about the genuine competence of the Authors about several chemical and physical aspects presented in the manuscript.
